# Cost-effectiveness of esketamine versus alternative treatment strategies for treatment-resistant depression in Hong Kong: A multi-armed modeling study

Yifan Li[1,2☯], Vivien Kin Yi Chan[2☯], Mark Jit[3,4,5], Franco Wing Tak Cheng[2,6], Hei Hang Edmund Yiu[2], David Makram Bishai[3], Dawn Craig[7], Esther Wai Yin Chan[2], Sandra Sau Man Chan[8*], Xue Li[1,2,9*]

**1** Department of Medicine, School of Clinical Medicine, Li Ka Shing Faculty of Medicine, The University of Hong Kong, Hong Kong SAR, China, **2** Centre for Safe Medication Practice and Research, Department of Pharmacology and Pharmacy, Li Ka Shing Faculty of Medicine, The University of Hong Kong, Hong Kong SAR, China, **3** School of Public Health, Li Ka Shing Faculty of Medicine, The University of Hong Kong, Hong Kong SAR, China, **4** Department of Infectious Disease Epidemiology, Faculty of Epidemiology and Population Health, London School of Hygiene & Tropical Medicine, London, United Kingdom, **5** Department of Global and Environmental Health, School of Global Public Health, New York University, New York, New York, United States of America, **6** Department of Pharmacy, The University of Hong Kong–Shenzhen Hospital, Hong Kong SAR, China, **7** Population Health Sciences Institute, Faculty of Medical Sciences, Newcastle University, Newcastle, United Kingdom, **8** Department of Psychiatry, Faculty of Medicine, The Chinese University of Hong Kong, Hong Kong SAR, China, **9** Department of Medicine, The University of Hong Kong–Shenzhen Hospital, Hong Kong SAR, China

☯ Co-first authors with equal contribution.

* schan@cuhk.edu.hk (SSMC); sxueli@hku.hk (XL)

## Abstract

### Background

Treatment-resistant depression (TRD), defined as failure to respond to at least two adequately administered antidepressant (AD) regimens, imposes major clinical and economic burdens. Esketamine nasal spray offers rapid antidepressant clinical effects, yet previous evaluations compared it only with unrealistic comparators such as AD monotherapy. This study assessed the cost-effectiveness of esketamine versus multiple alternative third-line strategies for TRD from the Hong Kong healthcare payer's perspective.

### Methods and findings

A Markov cohort model simulated adults with TRD in Hong Kong over 5 years with 4-week cycles. The model compared esketamine plus AD with six alternative third-line treatment strategies: combination therapy (AD plus AD), augmentation therapy (AD plus antipsychotic or lithium), psychotherapy alone, psychotherapy plus AD, repetitive transcranial magnetic stimulation (rTMS) plus AD, and electroconvulsive therapy (ECT) plus AD. Primary outcomes were quality-adjusted life-years

**Data availability statement:** The data used in this study cannot be shared publicly because the data custodian, the Hong Kong Hospital Authority (HA), which manages the Clinical Data Analysis and Reporting System (CDARS), has not granted permission for public release. Researchers who meet the criteria for access to confidential data may apply for access to CDARS data for research purposes through the Hospital Authority Data Sharing Portal (https://www3.ha.org.hk/data). The study protocol, detailed cost and efficacy parameters, and programming code used in this study are publicly available on GitHub (https://github.com/scan2030/scan2030-WP-3_TRD_ESK_CEA) and archived on Zenodo (https://doi.org/10.5281/zenodo.19520620).

**Funding:** This research was supported by the Hong Kong University Grants Committee Research Impact Fund (reference number: R7007-22 to XL). The University Grants Committee (UGC) is a public body that provides funding to support academic research in Hong Kong (https://www.ugc.edu.hk/eng/rgc/). The funders had no role in study design, data collection and analysis, decision to publish, or preparation of the manuscript.

**Competing interests:** I have read the journal's policy and the authors of this manuscript have the following competing interests: XL received research grants or contracts from the Health and Medical Research Fund (HMRF Main Scheme, HMRF Fellowship Scheme, and Hong Kong Special Administrative Region), and from the Research Grants Council Early Career Scheme (HKSAR); is also the former nonexecutive director of ADAMS Hong Kong; received commission grants from Hospital Authority of Hong Kong, internal funding from the University of Hong Kong, and research or education grants from Pfizer, Janssen and Bristol Myers Squibb (BMS); received consultancy fees from Merck Sharp & Dohme, Pfizer, Open Health, and The Office of Health Economics; and received honoraria for associate editorship from Nature Springer. HHEY reported receiving research grants from the Health Bureau of the Government of the Hong Kong SAR (HMRF) and Viatris, outside the submitted work.

(QALYs) and incremental cost-effectiveness ratios (ICERs) under a US$50,000/QALY willingness-to-pay (WTP) threshold. Deterministic and probabilistic sensitivity analyses and scenario analyses were conducted, focusing on alternative esketamine dosing, delivery strategies, and comparisons with other treatment options to assess the robustness of the results. In base-case analysis, esketamine was not cost-effective versus augmentation, combination, psychotherapy, or psychotherapy plus AD with ICERs ranging from US$134,127 to US$312,750 per QALY but was more cost-effective than rTMS (dominated) and ECT (ICER: US$322,407/QALY). Combination therapy was the most cost-effective among all strategies evaluated. The main limitation of this study is the reliance on indirect comparisons and assumptions derived from heterogeneous clinical trial populations, which may not fully reflect real-world patient characteristics and treatment pathways.

## Conclusions

Esketamine appeared more cost-effective than rTMS and ECT, but not cost-effective compared with other commonly used third-line treatment strategies for TRD. These findings suggest that cost-effectiveness evidence may help inform more context-sensitive treatment sequencing strategies beyond conventional line-of-therapy frameworks. Policy approaches such as price negotiation, optimized service delivery, and alternative dosing strategies may improve the value of esketamine for TRD management.

## Author summary

### Why was this study done?

- Treatment-resistant depression (TRD) is common and places a substantial burden on patients and healthcare systems.

- Esketamine is a novel and fast-acting treatment, but its economic value compared with treatments used in routine clinical practice remains unclear.

- Previous economic studies mainly compared esketamine with less realistic alternatives, rather than with the broader range of treatment strategies commonly used in clinical care.

### What did the researchers do and find?

- We developed a cost-effectiveness model to compare esketamine with six commonly used third-line treatment strategies for adults with TRD in Hong Kong over a 5-year period.

- We found that esketamine was more economically favorable than neurostimulation therapies such as repetitive transcranial magnetic stimulation (rTMS) and electroconvulsive therapy (ECT), but was not cost-effective compared with

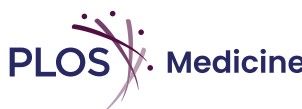

**Abbreviations:** AD, antidepressant; BMS, Bristol Myers Squibb; CBT, Cognitive Behavioral Therapy; CDARS, Clinical Data Analysis and Reporting System; CEACs, cost-effectiveness acceptability curves; DBT, Dialectical Behavior Therapy; DSA, deterministic sensitivity analyses; ECT, electroconvulsive therapy; EMAEuropean Medicines AgencyEMR, electronic medical records; FDA, Food and Drug Administration; HA, Hospital Authority; HKD, Hong Kong Dollars; HRQoL, health-related quality of life; ICERs, incremental cost-effectiveness ratios; IPT, Interpersonal Therapy; IV ketamine, intravenous ketamine; MADRS, Montgomery–Åsberg Depression Rating Scale; NICE, National Institute for Health and Care Excellence; PSA, probabilistic sensitivity analyses; QALYs, quality-adjusted life years; rTMS, repetitive transcranial magnetic stimulation; RRs, relative risks; TRD, treatment-resistant depression; UGC, University Grants Committee; WTP, willingness-to-pay.

commonly used strategies such as combination therapy, augmentation therapy, or psychotherapy-based treatments at standard decision thresholds.

- Among all strategies, combination therapy was the most cost-effective option, and the main findings remained broadly similar when we tested different assumptions in the model.

### What do these findings mean?

- Although esketamine offers clinical benefits, its high cost currently limits its economic value compared with other widely used treatments, and improving its cost-effectiveness may require price reductions or more efficient service delivery.

- These findings may help inform clinical guidelines and healthcare policy decisions by supporting more cost-effective treatment sequencing strategies.

- This study is limited by its use of a modeling approach and reliance on data from multiple sources rather than direct patient comparisons.

## 1. Introduction

Although oral antidepressants (ADs) are important treatment modalities for managing depression, over half of patients do not respond to the first course of AD, and a significant proportion remain nonresponsive to subsequent lines of treatment [1]. This condition is "treatment-resistant depression (TRD)"; a source of immense burden in mortality, morbidity, and economic cost associated with patients with depression. Patients with TRD have a 29%–39% higher risk of all-cause mortality compared to those without TRD. In addition to the elevated risk of self-harm, TRD is also associated with the development of various physical comorbidities—particularly cardiovascular disease and stroke—which further increase mortality. Moreover, TRD is closely linked to higher direct medical costs, productivity losses, and employment changes [2,3]. Pharmaceutical strategies to manage TRD commonly include switching between oral ADs, combining oral ADs, or augmenting current therapy with antipsychotics [4,5]. However, they are unlikely to improve outcome if patients are pharmacologically nonresponsive to the existing drug classes within oral ADs.

 Esketamine nasal spray was first approved by the U.S. Food and Drug Administration (FDA) and the European Medicines Agency (EMA) in 2019 for treating adults with TRD in combination with an oral AD [6,7]. As a novel antidepressant with rapid onset and a distinct mechanism of action, esketamine has received considerable policy interest and clinical attention internationally. To date, esketamine has been approved in 77 countries and used by over 140,000 patients worldwide [8]. Distinct from conventional oral ADs, esketamine is an N-methyl-D-aspartate receptor antagonist that mainly upregulates the release of brain-derived neurotrophic factor, a crucial substance restoring synaptic remodeling which takes weeks to produce with oral ADs [9,10]. The intranasal route of administration further contributes to its rapid onset of symptom relief. Compared

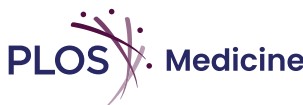

with oral AD monotherapy, Phase III trials show that esketamine in addition to an oral AD significantly reduced symptoms of depression (measured using the Montgomery–Åsberg Depression Rating Scale (MADRS) score) after 4 weeks, with immediate change as early as the second day of treatment which continued for 28 days, in contrast to 2–4 weeks with oral ADs [11]. The same trials found that continued treatment with esketamine among responding patients prolonged the median time until relapse by 7.5 weeks among the full population in each arm with a relapse rate of 26.7% (45.3% in the control) [12].

Although efficacy is favorable, esketamine currently has a high market price. Ross and colleagues found that in the United States esketamine as adjunct to oral AD compared to oral AD alone was unlikely to be cost-effective, unless the price fell by 40% [13]. Five other modeling studies in the United States, Italy, and Canada gave similar conclusions when comparing esketamine to oral AD, intravenous ketamine, and electroconvulsive therapy (ECT) [14–18]. Given the reported price up to US$5,000 per month [13], the health and economic benefits could be reasonably outweighed by the additional cost. Unlike most pharmacological treatments, patients must be supervised by healthcare professionals during drug administration, which further raises the cost for clinical monitoring.

However, the existing cost-effectiveness studies had limitations. Most models compared adjunctive esketamine to oral AD alone, but the latter is not a realistic comparator. Based on clinical guidelines, treatment pathway research and technical appraisal documents by the National Institute for Health and Care Excellence (NICE) in the United Kingdom, oral AD monotherapy was not the mainstay of treatment among depression patients who had already failed two lines of treatment [2,4,5,19]. Of all real-world drug treatments for TRD in the Hong Kong setting, oral AD monotherapy accounts for only 42% of choices, as patients tend to have combination/augmentation therapies (42%) or polytherapy (16%) [2]. Nonpharmaceutical choices such as psychotherapy and neurostimulation therapies (e.g., rTMS and ECT) are also common interventions for TRD. One reason for solely comparing to oral AD monotherapy was the heavy reliance on Phase III esketamine trial data, which did not apply the usual real-world treatments for TRD as the control. Options other than oral AD monotherapy were therefore modeled inadequately but still of economic interest.

Motivated by the clinical and policy interest in esketamine as a novel and rapid-acting antidepressant, this study aims to compare esketamine nasal spray with six commonly used real-world treatment options to provide a comprehensive economic evaluation of third-line strategies for TRD management from the healthcare payer's perspective in Hong Kong. In accordance with the NICE technology appraisal discussion (Sections 3.3 and 3.6) [19], we chose augmentation therapy as the reference comparator. In real-world clinical practice, patients with TRD are typically offered augmentation therapy following two unsuccessful antidepressant treatments. By systematically evaluating multiple real-world comparators, this study not only assesses the relative economic value of esketamine but also seeks to identify clinically relevant and cost-effective treatment strategies.

## 2. Methods

This study was conducted in accordance with the Consolidated Health Economic Evaluation Reporting Standards (CHEERS 2022) checklist [20]. The study protocol was approved by the Institutional Review Board of the University of Hong Kong/Hospital Authority Hong Kong West Cluster (HKU/HA HKW IRB) (Approval No.: UW 20-218). The requirement for informed consent was waived due to the use of fully anonymized data. All model inputs, assumptions, and results were cross-checked by two independent researchers (YL and VKYC) to ensure quality, accuracy, and transparency.

### 2.1. Data sources

To better generalize the modeled outcomes to real-world economic and clinical contexts, we used the territory-wide electronic medical records (EMR) database managed by the Hong Kong Hospital Authority as the primary source of model inputs. The database keeps records of patients accessing publicly funded healthcare services in Hong Kong including demographic attributes, deaths, attendances, diagnoses in International Classification of Diseases, 9th Revision, Clinical Modification (ICD-9-CM) codes and prescriptions in all service settings. The anonymous data covers 90% of hospital admissions and 76% of chronic care among over 7.4 million eligible residents [21,22]. In this study, the demographic characteristics, background costs

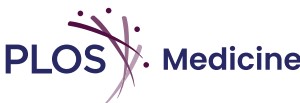

of healthcare resource utilization and mortality rates were derived from a reference cohort in this EMR database. Although it provides detailed healthcare use and cost data, clinical outcomes such as depression severity scores, treatment response, or remission status are not included. Therefore, we did not estimate clinical effectiveness directly from real-world data but used published clinical trials and meta-analyses to inform treatment efficacy and transition probabilities in the model.

## 2.2. Interventions

We evaluated seven treatment strategies in our cost-effectiveness model: (i) augmentation of an oral AD with either an antipsychotic (olanzapine, cariprazine, quetiapine, or ziprasidone) or lithium (reference group); (ii) esketamine adjunctive to an oral AD; (iii) combining an oral AD with another oral AD (mianserin or mirtazapine); (iv) psychotherapy alone (Cognitive Behavioral Therapy [CBT], Dialectical Behavior Therapy [DBT], or Interpersonal Therapy [IPT]); (v) any psychotherapy adjunctive to oral AD; (vi) unilateral repetitive transcranial magnetic stimulation (rTMS) adjunctive to oral AD; and (vii) electroconvulsive therapy (ECT) adjunctive to oral AD. These strategies serve as the third-line treatment for depression based on prescribing patterns observed in real-world clinical practice.

## 2.3. Model description

We adapted a published state-transition model commonly used for economic evaluations for antidepressants (Fig 1) [13,17]. The model consists of at least three treatment lines with five health states per line, including: Treatment,

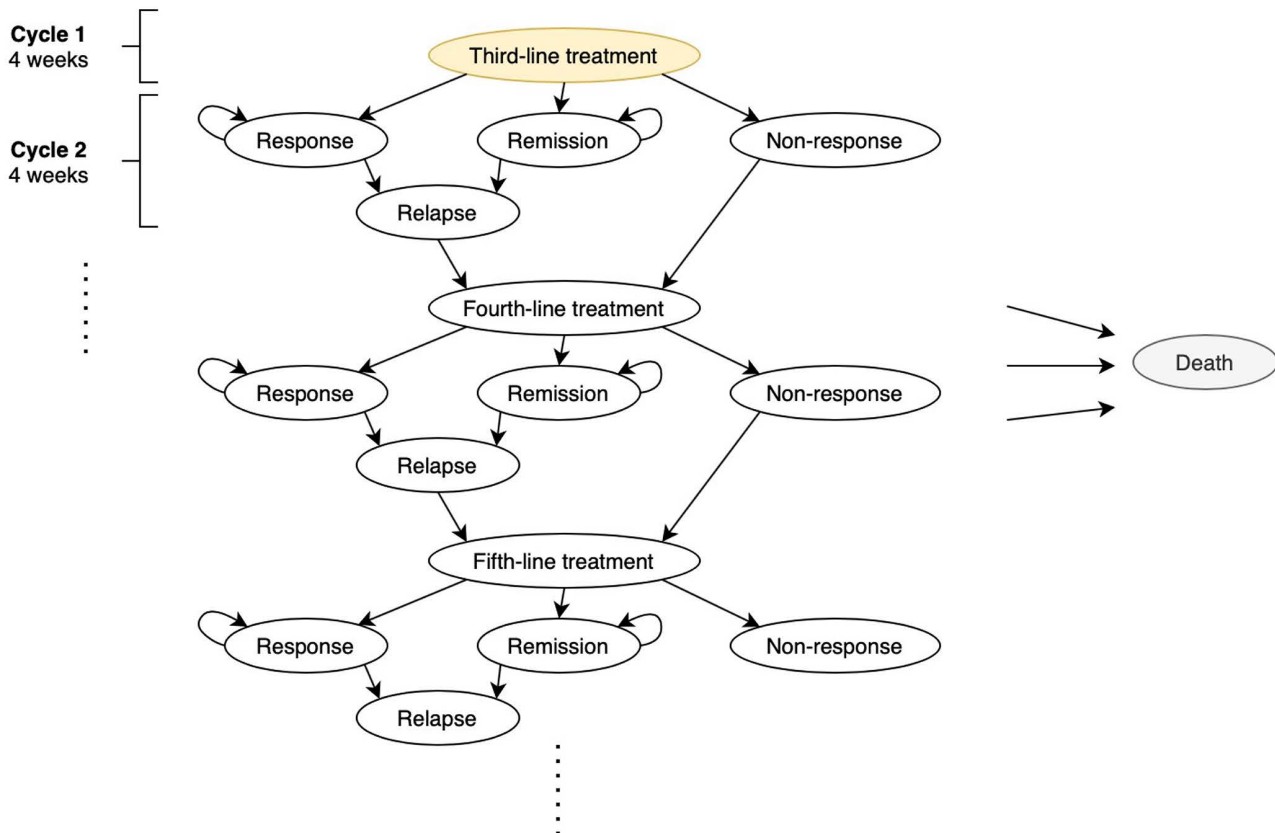

**Fig 1. Schema of Markov model structure.**

Response, Remission, Nonresponse, and Relapse. Upon ascertainment as TRD, patients were initiated on the first treatment (equivalent to the third-line therapy since depression diagnosis) as specified in Section 2.2 for 4 weeks before proceeding to other states in the next cycle based on clinical improvement. "Response" frequently refers to at least a 50% reduction in depression symptom scores, whilst "remission" refers to achieving a score below an absolute threshold originally defined by the sourced articles which may differ between the choice of measuring instruments. "Nonresponse" commonly refers to a state not responding nor remitting to treatment, and "relapse" is the recurrence of symptoms after response and remission when the absolute score exceeds a threshold pre-defined in the scales. When patients experienced a relapse or nonresponse, they would initiate a next-line treatment in the ensuing cycle. Lastly, all patients in any states may proceed to all-cause mortality (absorbing state).

### 2.4. Model inputs

Key model inputs, including distributional assumptions, sensitivity analysis ranges, and data sources, are detailed in Table A in S1 Appendix. Demographic characteristics were derived from Hong Kong EMRs between 2014 and 2016, identifying patients diagnosed with TRD [3,23]. The target population was a hypothetical cohort of adult patients (aged ≥18 years) with TRD. TRD was identified in the EMR as patients who had at least two prior antidepressant switches, followed by a third-line treatment regimen prescribed for at least 4 weeks. Mortality rates were adapted from a previously developed burden projection model specific to the TRD population [23]. Clinical effectiveness parameters were informed by meta-analyses and randomized controlled trials using oral AD monotherapy as a common comparator to facilitate indirect comparisons across treatment strategies. Remission, response, and relapse probabilities were specified by treatment line to reflect increasing treatment resistance, and were aligned with STAR*D trial levels 2, 3, and 4 for third-line, fourth-line, and fifth-line treatments, respectively [1]. To ensure consistency with the model's 28-day cycle length, all annual rates were converted to per-cycle probabilities using the formula $p = 1 - e^{[\ln(1-r)/t]*4}$ for $t$ weeks of follow-up [24]. Efficacy estimates for esketamine were drawn from the TRANSFORM-1/2 and SUSTAIN-1 trials [11, 12, 25], while data for other treatment strategies were extracted from Cochrane systematic reviews [26,27] and multiple meta-analyses [28–35]. Treatment costs were based on the internal healthcare utilization data, published studies [ 35–43], and Hospital Authority's official fee schedules [44,45]. Treatment durations and healthcare visit frequencies were determined from trial protocols, clinical guidelines, and expert opinion derived through consensus discussions among a consultant psychiatrist, a health economist, and a clinical pharmacist, to reflect real-world practice in the Hong Kong public healthcare setting [25,46–51]. Health utility values for different health states were sourced from Sapin and colleagues, who assessed health-related quality of life (HRQoL) among subgroups of patients with TRD [52]. A detailed description of parameter assumptions and data sources is provided in Supporting Methods in S1 Appendix.

### 2.5. Statistical analysis

#### 2.5.1. Base-case analysis.

We constructed a cohort Markov model using R (Version R 4.4.1) and Microsoft Excel (Version 16) to project the expected costs and health benefits of esketamine and six other third-line treatment choices commonly found in clinical practice. The primary outcome was the ICER calculated as cost per quality-adjusted life-year (QALY) gained. A willingness-to-pay (WTP) threshold of US$50,000 per QALY (approximately one times Hong Kong GDP per capita) was applied to assess cost-effectiveness [53]. A 5-year time horizon was adopted to allow sufficient accrual of long-term costs while minimizing uncertainty from extrapolated effectiveness from the Hong Kong public healthcare payer's perspective, which was chosen to reflect resource allocation and reimbursement decisions within the publicly funded healthcare system. This period corresponds to the maximum follow-up possible since esketamine's initial approval in Hong Kong in 2019 and encompasses the duration of current clinical trial data (up to 24 months). The cycle length was 4 weeks, corresponding to the minimum recommended duration to assess responsiveness to an intervention for

depression [4]. All costs and health benefits were discounted at 3% per annum, in line with commonly adopted practice in cost-effectiveness analyses and recent evaluations conducted from a Hong Kong public healthcare payer perspective [54]. Cost inputs were expressed in 2024 US Dollars (US$). Costs originally reported in Hong Kong Dollars (HKD) were converted using at an exchange rate of 1.00 to 7.80 HKD.

**2.5.2. Scenario and sensitivity analyses.** To explore uncertainties around real-world esketamine use, eight scenario analyses were conducted: (1) reducing dosing frequency from once weekly to biweekly after week 9, (2) extending the treatment cycle length from 4 to 8 weeks for the esketamine arm, implemented by adopting an 8-week model cycle to reflect a longer clinical follow-up interval, (3) substituting Specialist Outpatient Clinic and psychiatric day hospital costs with direct intranasal administration and monitoring costs, (4) reducing each esketamine dose from 84 mg to 56 mg, (5) applying a 3.5% annual discount rate to both costs and health outcomes, in line with UK NICE reference-case guidance, (6) assuming a 75% reduction in the esketamine acquisition price to reflect potential future price negotiations or generic entry, (7) extending the analytic time horizon from 5 years to a lifetime horizon (20 years) to assess the impact of truncating long-term costs and outcomes in the base-case analysis, and (8) estimating the total esketamine arm cost required for esketamine to achieve cost-effectiveness at a WTP threshold of US$50,000 per QALY, assuming unchanged effectiveness and linear cost scaling. One-way deterministic sensitivity analyses (DSA) were conducted to examine the impact of key parameters on the ICER of esketamine relative to each of the other six treatment strategies. Each parameter was varied independently to its upper and lower bounds while holding others constant. Ranges were defined using 95% confidence intervals or derived through parametric bootstrapping. Probabilistic sensitivity analyses (PSA) were performed using 10,000 Monte Carlo simulations to simultaneously sample all parameters from assigned distributions. For each WTP threshold, we computed net monetary benefit (NMB = WTP × ΔQALY − ΔCost) and plotted cost-effectiveness acceptability curves (CEACs) to illustrate the probability of each strategy being cost-effective.

## 3. Results

### 3.1. Base-case analysis

Table 1 presents the results of the base-case cost-effectiveness analysis per patient. Under a WTP threshold of US$50,000/QALY, esketamine was not cost-effective when compared to augmentation therapy, combination therapy, psychotherapy alone, and psychotherapy plus AD, with incremental cost-effectiveness ratios (ICERs) ranging from US$134,127 to US$312,750 per QALY. In contrast, esketamine dominated rTMS, being both less costly and more effective. Compared with esketamine, ECT was associated with a high ICER (US$322,407/QALY), exceeding the WTP threshold. Combination therapy was the most cost-effective strategy, with all other treatment options having higher ICERs when compared against the reference group (AUG).

The solid gray line in Fig 2 connects the nondominated strategies which forms the efficiency frontier. Psychotherapy alone and rTMS adjunctive to an oral AD fall above this frontier, an indication that they are dominated strategies, being more costly and less effective than at least one alternative. On the cost-effectiveness frontier, combination therapy was the least costly nondominated strategy, followed by psychotherapy plus antidepressant therapy (PSY + AD), esketamine plus antidepressant therapy (ESK + AD), and electroconvulsive therapy plus antidepressant therapy (ECT + AD). The ICER for ESK + AD compared with the next least costly nondominated alternative (PSY + AD) was US$312,750 per QALY. Moving further along the frontier, the ICER for ECT + AD compared with ESK + AD was US$322,407 per QALY.

### 3.2. Scenario analysis

Scenario analysis results are presented in Table B in S1 Appendix. Except for Scenario 5, all other esketamine-related scenarios yielded lower ICERs than the base-case analysis, indicating improved cost-effectiveness relative to the base-case assumptions. Scenario 8 was explicitly designed as a cost-threshold analysis. Under this scenario, holding health

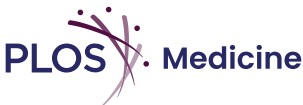

**Table 1. Base-case analysis of per patient cost, QALYs and ICER in each comparative arm.**

| Treatment Strategy | Cost (US$) | QALYs | Incremental Cost (US$,vs. Ref) | Incremental QALYs(vs. Ref) | ICER: other strategies vs. Ref (AUG) (US$/QALY) | ICER: ESK+AD vs. other strategies (US$/QALY) | ICER: vs. the next nondominated strategy (US$/QALY) |
|---|---|---|---|---|---|---|---|
| AUG (Ref) | 16,185 | 2.895 | Ref | Ref | – | 234,109 | Dominated* |
| COM | 16,163 | 2.903 | −22 | 0.008 | Dominant* | 274,426 | Ref |
| PSY alone | 19,538 | 2.879 | 3,353 | −0.016 | Dominated* | 134,127 | Dominated* |
| PSY+AD | 21,555 | 2.926 | 5,370 | 0.031 | 173,226 | 312,750 | 234,435 (vs COM) |
| ESK+AD | 29,061 | 2.950 | 12,876 | 0.055 | 234,109 | – | 312,750 (vs PSY+AD) |
| rTMS+AD | 30,607 | 2.936 | 14,422 | 0.041 | 351,756 | Dominant* | Dominated* |
| ECT+AD | 46,471 | 3.004 | 30,286 | 0.109 | 277,853 | †322,407 | 322,407 (vs ESK+AD) |

**Notes:** 1. Dominant*: a strategy is associated with lower costs and higher effectiveness (QALYs) compared with the comparator.

2. Dominated*: a strategy is associated with higher costs and lower or equal effectiveness. Dominated strategies were not considered cost-effective and were therefore not assigned an ICER versus the next nondominated strategy.3. 322,407†: the ICER of US$322,407 per QALY corresponds to the comparison of ECT+AD versus ESK+AD, representing the additional cost required to gain one extra QALY when moving from ESK+AD to ECT+AD.4. Abbreviations: Ref, Reference Comparator; AUG, Augmentation therapy (antidepressant combined with antipsychotic/lithium); COM, Combination therapy (antidepressant combined with antidepressant); PSY alone, Psychotherapy alone; PSY+AD, Psychotherapy combined with antidepressant; ESK+AD, Esketamine combined with antidepressant; rTMS+AD, Repetitive transcranial magnetic stimulation combined with antidepressant; ECT+AD, Electroconvulsive therapy combined with antidepressant.

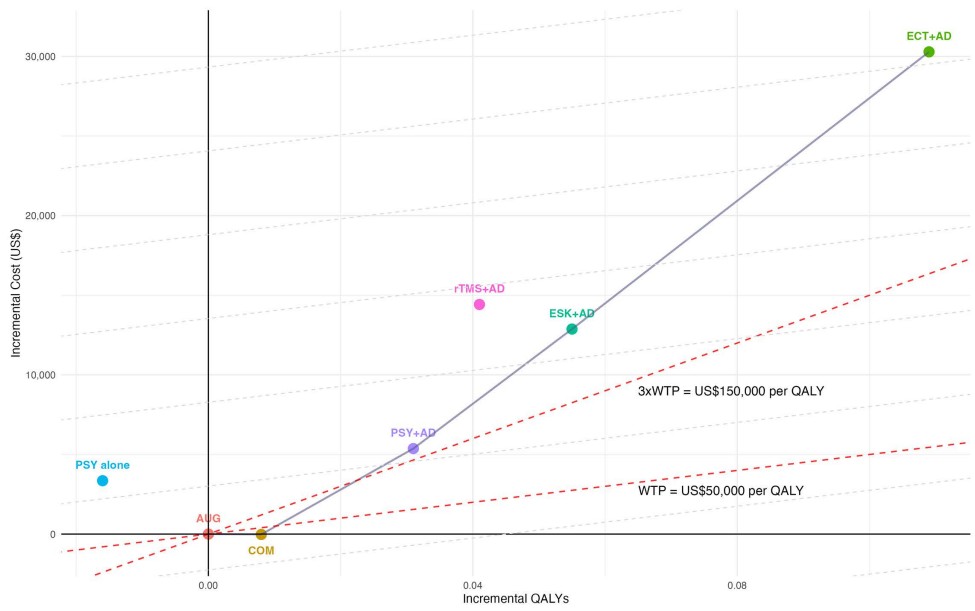

**Fig 2. Cost-effectiveness efficiency frontier of treatment-resistant depression (TRD) strategies.** Each point represents a third-line treatment strategy for TRD compared with augmentation therapy. Red dashed lines indicate willingness-to-pay (WTP) thresholds of US$50,000 and US$150,000 per quality-adjusted life-year (QALY). The solid gray line represents the efficiency frontier formed by nondominated strategies. Gray dashed lines represent indifference curves. Strategies located below and to the left of a curve are considered more cost-effective than those above and to the right. Abbreviations: WTP, willingness-to-pay; QALY, quality-adjusted life-year; AUG, Augmentation therapy (antidepressant combined with antipsychotic/lithium); COM, Combination therapy (antidepressant combined with antidepressant); PSY alone, Psychotherapy alone; PSY+AD, Psychotherapy combined with antidepressant; ESK+AD, Esketamine combined with antidepressant; rTMS+AD, Repetitive transcranial magnetic stimulation combined with antidepressant; ECT+AD, Electroconvulsive therapy combined with antidepressant.

outcomes constant, a reduction in total per patient costs to US$18,935 (approximately a 35% reduction from the base-case) resulted in an ICER of US$50,000 per QALY. Importantly, apart from this threshold-based scenario, esketamine did not become cost-effective at a WTP threshold of US$50,000 per QALY over the 5-year time horizon in any of other scenario analyses. Among the scenarios evaluated, Scenario 2, which extended the model cycle length from 4 to 8 weeks to reflect a longer clinical follow-up interval, yielded the most favorable ICER (US$73,556/QALY). This result reflects a modeled structural scenario rather than a direct change in treatment efficacy. For Scenario 6, which applied a 75% reduction in the acquisition cost of esketamine, the ICER relative to augmentation decreased substantially to US$116,327 per QALY, falling within three times the WTP threshold (US$150,000/QALY). Scenario 7 extended the analytic horizon from 5 to 20 years. Over this longer time horizon, esketamine accrued total costs of US$63,473 and 9.05 QALYs. Compared with augmentation over the same 20-year horizon, this corresponded to an ICER of US$229,750 per QALY, indicating that esketamine remained not cost-effective.

### 3.3. Sensitivity analyses

The top 10 influential parameters for each comparison are displayed in tornado diagrams for the DSA (Fig B in S1 Appendix). Across all comparisons, model parameters related to the efficacy of esketamine—particularly the relative risks (RRs) of transitioning from remission to relapse and from treatment to remission—exerted the greatest influence on ICER values. In the comparison between esketamine and augmentation therapy (reference group), varying the RR of remission to relapse for the esketamine strategy led to ICER fall below three times GDP (US$150,000/QALY). Despite this wide variation, the cost-effectiveness conclusion remains unchanged.

PSA results in Fig 3 show that while most PSA simulations for esketamine exceed the US$50,000/QALY threshold, its cost-effectiveness is generally more favorable than ECT and rTMS. These findings are consistent with the CEACs (Fig 4). At a WTP of US$50,000/QALY, combination therapy had the highest probability of cost-effectiveness (75%), followed by augmentation therapy (25%), while esketamine was not cost-effective (0%). Even at US$150,000/QALY, esketamine remained less likely to be cost-effective (15.1%) compared to combination (58.3%) and augmentation therapy (15.6%). Esketamine only surpassed other strategies in cost-effectiveness probability when the WTP threshold exceeded US$213,675/QALY—more than four times the base-case threshold.

## 4. Discussion

In this study, we compared esketamine nasal spray with six commonly used third-line treatment strategies for TRD and found that its economic value varied substantially across different comparators. Unlike previous studies that primarily compared esketamine with oral AD monotherapy or only single comparator, our model incorporates a broader spectrum of both pharmacological and nonpharmacological comparators, better reflecting actual clinical practice. Beyond assessing the value of esketamine, our analysis also identifies the most cost-effective treatment option relative to the realistic multiple alternative comparator therapies. Notably, our study improves upon earlier models by leveraging a territory-wide real-world database, rather than relying solely on international trial-based evidence. This design enhances both the contextual relevance and the generalizability of our findings to routine clinical settings.

The base-case results indicated that esketamine was not cost-effective compared to most other treatment strategies under the WTP threshold of US$50,000 per QALY, except when compared with rTMS and ECT. Among all therapies evaluated, combination therapy emerged as the most cost-effective option. Our results are broadly consistent with prior economic evaluations of esketamine conducted in the United States, Canada, and Europe. For example, Ross and colleagues reported base-case ICERs of approximately US$237,000–242,000/QALY from both societal and healthcare sector perspectives in the United States, suggesting that esketamine is unlikely to be cost-effective at conventional thresholds. They also indicated that esketamine would require substantial price reductions (>40%) to reach acceptable

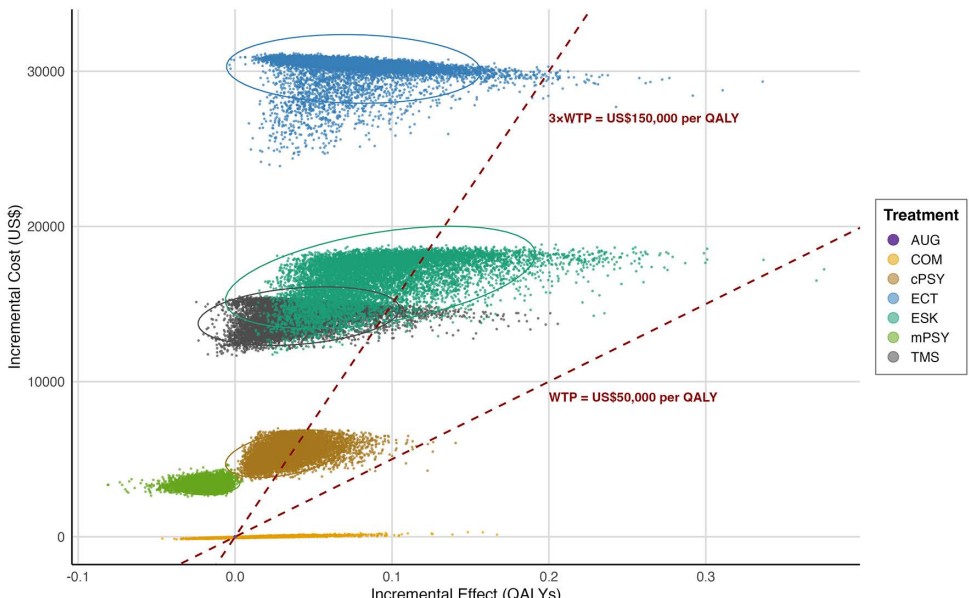

**Fig 3. Probabilistic sensitivity analysis results (10,000 Monte Carlo simulations) on the incremental cost-effectiveness scatter plot.** Each dot represents one simulation. Ellipses indicate 95% uncertainty intervals. The dashed lines represent willingness-to-pay (WTP) thresholds of US$50,000 and US$150,000 per quality-adjusted life-year (QALY). Abbreviations: AUG, augmentation therapy; COM, antidepressant combination therapy; cPSY, psychotherapy alone; mPSY, psychotherapy combined with antidepressant; ESK, esketamine combined with antidepressant; TMS, repetitive transcranial magnetic stimulation combined with antidepressant; ECT, electroconvulsive therapy combined with antidepressant; WTP, willingness-to-pay; QALY, quality-adjusted life-year.

cost-effectiveness [13]. Similar conclusions were drawn by Canada's CADTH report, where esketamine's ICER was approximately US$125,376/QALY, necessitating an estimated 60% price reduction to achieve cost-effectiveness at a WTP of US$50,000/QALY [18]. Likewise, Rognoni and colleagues found esketamine was not cost-effective from the Italian healthcare system perspective but could be justified from a societal perspective, primarily due to productivity gains from rapid functional improvement and reduced work loss [17].

However, our finding that esketamine appears more cost-effective than ECT contrasts with the previous economic evaluation in which ECT was found to be cost-effective or even dominant over esketamine [16]. This discrepancy is not unexpected and can be largely explained by two key methodological differences: cost structure assumptions and effectiveness modeling choices.

First, substantial differences exist in how ECT costs are defined across studies. In our analysis, the unit cost of ECT was based on the Hospital Authority public charge for noneligible persons in Hong Kong, reflecting the full, unsubsidized procedural cost, including anesthesiology, psychiatric consultation, nursing labor, recovery facilities, and institutional overheads. In contrast, this previous study derived ECT costs from NICE-based bundled tariffs (approximately £558 per session), which represent subsidized average payments within the UK National Health Service rather than disaggregated, labor-intensive costs. Given that ECT is predominantly labor-driven, such differences in healthcare financing and labor cost structures can substantially influence total cost estimates and downstream cost-effectiveness results. This interpretation is further supported by our DSA (Fig B in S1 Appendix). When ECT-related procedural and inpatient admission costs were varied by ±50%, the incremental cost-effectiveness ratio for ECT versus esketamine changed substantially, confirming that comparative cost-effectiveness results are highly sensitive to labor- and procedure-related cost assumptions. Although reductions in ECT costs markedly lowered the ICER, ECT did not become cost-effective relative to esketamine

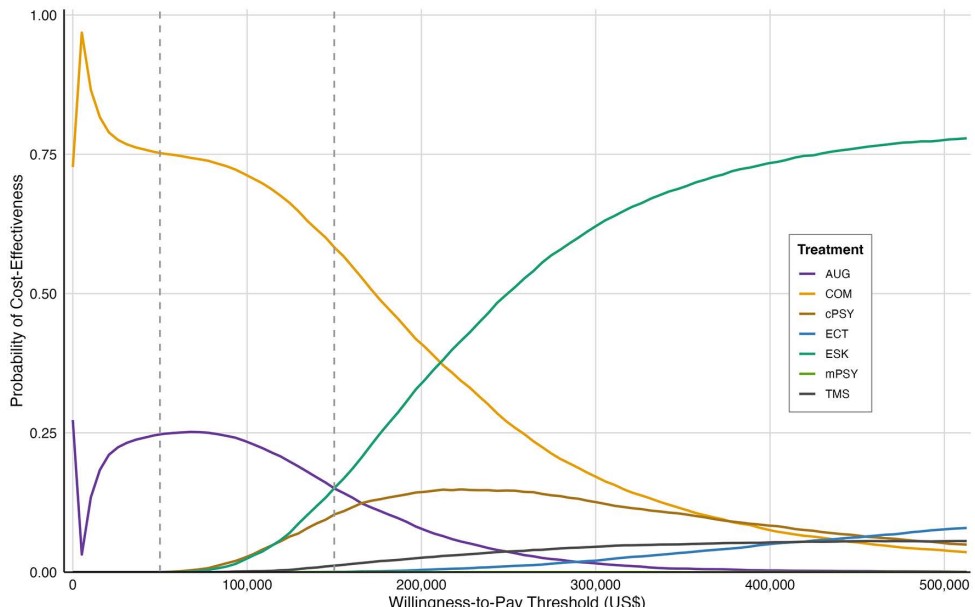

**Fig 4. Cost-effectiveness acceptability curves (CEACs) based on 10,000 Monte Carlo simulations.** Curves show the probability that each treatment strategy is the most cost-effective across a range of willingness-to-pay (WTP) thresholds. Gray dashed lines indicate the base-case WTP threshold of US$50,000 per quality-adjusted life-year (QALY) and the three times GDP per capita threshold of US$150,000 per QALY. The threshold at which esketamine first becomes the most cost-effective option is US$213,675 per QALY. Abbreviations: CEAC, cost-effectiveness acceptability curve; WTP, willingness-to-pay; QALY, quality-adjusted life-year; AUG, augmentation therapy; COM, antidepressant combination therapy; cPSY, psychotherapy alone; mPSY, psychotherapy combined with antidepressant; ESK, esketamine combined with antidepressant; TMS, repetitive transcranial magnetic stimulation combined with antidepressant; ECT, electroconvulsive therapy combined with antidepressant.

at the base-case WTP threshold of US$50,000/QALY in any of the scenarios examined. Together, these findings underscore that differences in healthcare financing structures and labor cost accounting can materially affect cost-effectiveness estimates for labor-intensive interventions such as ECT, and highlight the importance of context-specific costing in economic evaluations.

Second, differences in modeled effectiveness assumptions may further contribute to divergent findings. In our model, ECT efficacy parameters were primarily sourced from the broader TRD literature and were not consistently restricted to the most severe or urgent clinical subgroups for whom ECT is typically indicated in real-world practice. As a result, the incremental QALY gains associated with ECT may be conservatively estimated. If ECT effectiveness were higher among patients with more severe, refractory, or life-threatening depression as suggested in some clinical studies, its true QALY benefit could exceed that captured in our base-case analysis, thereby improving its cost-effectiveness profile. This interpretation is supported by our DSA (Fig B in S1 Appendix). When the relapse risk from remission under ECT was reduced (relative risk of remission-to-relapse = 0.45), the ICER for esketamine versus ECT decreased substantially to approximately US$191,000 per QALY, approaching three times the conventional WTP threshold. This scenario reflects a clinical context in which ECT is more effective at sustaining remission, as may be expected among more severe or highly refractory patient populations. Although ECT did not become cost-effective relative to esketamine even under this more optimistic effectiveness assumption, the marked reduction in the ICER illustrates that ECT cost-effectiveness is highly sensitive to assumptions regarding relapse prevention and patient severity. Therefore, the combination of higher modeled ECT costs and potentially conservative assumptions regarding ECT effectiveness in our analysis helps explain why esketamine appears economically favorable in the Hong Kong–specific setting, whereas studies conducted in lower-labor-cost or differently subsidized healthcare systems have reported contrasting conclusions.

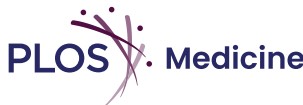

Taken together, these considerations also have important implications for the generalizability of our findings. Our findings should be interpreted in the context of a high-income healthcare system with relatively high labor costs. Although esketamine appeared economically favorable compared with rTMS and ECT in Hong Kong, this result is largely driven by the labor-intensive nature of rTMS and ECT, which require substantial physician, nursing, anesthetic, and facility resources. In healthcare systems with lower-labor-costs, the relative cost-effectiveness of these interventions may differ substantially, even if clinical effectiveness remains unchanged. This is particularly relevant for other labor-intensive or service-dependent interventions. For example, treatments such as intravenous ketamine (IV ketamine) have cost structures that are highly dependent on administration, monitoring, and service delivery requirements, making their economic value sensitive to local implementation practices. The component-level cost reporting in this study enables context-specific re-estimation and supports adaptation of the model to alternative healthcare settings.

Moreover, sensitivity and scenario analyses did not materially alter the conclusions of the base-case results. DSA identified the efficacy parameters of esketamine—particularly the relative risks of transitioning from remission to relapse—as the most influential drivers of cost-effectiveness. However, even under the most favorable assumptions, the resulting ICER still remained above the WTP threshold. PSA further confirmed the limited cost-effectiveness of esketamine, with a 0% probability at the WTP threshold, and only a 15.1% probability at a threshold three times the WTP threshold (US$150,000/QALY). Esketamine was only observed to be the most economically favorable strategy when the WTP threshold exceeded approximately US$213,675/QALY. In Scenario 6, which examined reductions in the esketamine drug acquisition price alone, we found that a 75% price reduction was sufficient to reduce the ICER to below a higher WTP threshold corresponding to three times WTP (US$150,000/QALY), but remained above the conventional WTP (US$50,000/QALY). By contrast, achieving cost-effectiveness at the WTP threshold would require a substantially larger reduction in total esketamine arm costs, rather than drug price alone. Therefore, in Scenario 8, under highly optimistic structural assumptions—including linear drug pricing and a clear separation of drug and monitoring cost components—a reduction of ~35% in total esketamine-related costs would be required to approach a WTP threshold of US$50,000/QALY. At higher thresholds, the probability of cost-effectiveness increases, as shown in the CEACs (Fig 4).

While none of the esketamine scenarios achieved cost-effectiveness under the current threshold, scenario analysis offered valuable insights into potential strategies to improve its economic profile. Notably, Scenario 2, which extended the model cycle length from 4 to 8 weeks to reflect a longer clinical follow-up interval, resulted in a substantially lower ICER (US$73,556/QALY). Importantly, this improvement was driven by structural features of the model. Specifically, fewer transition opportunities per year and prolonged time spent in response and remission states, rather than by any assumed increase in treatment efficacy. Therefore, this scenario should be interpreted as an exploratory, upper-bound structural scenario rather than as evidence that less frequent clinical monitoring itself improves health outcomes. Although existing clinical guidelines often recommend follow-up intervals of one to two months in stable patients, the findings from Scenario 2 do not imply a causal benefit of reduced monitoring frequency, but instead illustrate how cost-effectiveness estimates may be sensitive to assumptions regarding cycle length and transition timing. Additionally, substituting specialist outpatient clinic and psychiatric day hospital services with simplified nurse-led monitoring and intranasal protocols substantially reduced overall costs. These results imply that apart from considerations around drug pricing, policy interventions aimed at optimizing care delivery may also improve the cost-effectiveness of esketamine to some extent.

Despite esketamine's current limited economic appeal, it remains a potentially valuable option due to the rapid onset of action and relapse prevention benefits for patients with severe TRD who have not responded to other therapies. However, routine use may not be economically viable at existing price and reimbursement levels. In contrast, our findings importantly highlight that combination therapy emerged as the most cost-effective third-line strategy, yet it remains underutilized in real-world clinical practice. More broadly, our results suggest that future economic evaluations may benefit from more focused comparisons between clinically similar third-line interventions, such as esketamine and rTMS. Because these treatments are often used in overlapping patient populations but differ fundamentally in cost structure (drug-intensive

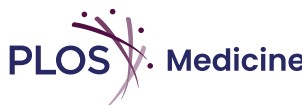

versus labor-intensive), such comparisons may yield more transferable and methodologically robust insights across healthcare systems. Policymakers could consider increasing awareness and promoting wider adoption of combination therapy, as this would likely yield substantial economic and clinical benefits. For esketamine, strategies such as price negotiations or conditional reimbursement schemes could enhance its cost-effectiveness, given the substantial quality of life improvements associated with its use. Additionally, health authorities could also consider applying higher WTP thresholds to economic evaluations. More broadly, findings from comparative cost-effectiveness analyses may help inform more comprehensive and context-sensitive treatment sequencing strategies for TRD, beyond conventional line-of-therapy frameworks. Such evidence may support clinical decision-making and resource allocation in later-line treatment settings.

This study has several limitations. First, in the absence of head-to-head trials directly comparing esketamine with other real-world treatment strategies, our analysis relied on anchored indirect comparisons using antidepressant monotherapy as a common reference. While this approach allows internally consistent comparisons across treatment arms within the model framework, the effectiveness estimates were drawn from clinical trials and meta-analyses conducted in populations that may differ in baseline disease severity, degree of treatment resistance, and definitions of response or remission. As a result, the comparative effectiveness estimates may be subject to residual confounding arising from heterogeneity across evidence sources and should be interpreted with caution. In particular, some interventions, such as ECT, are typically reserved for a highly selected subgroup of patients with severe or refractory depression, applying average treatment effects across the broader TRD population may underestimate its health benefits for appropriate candidates. Consequently, the cost-effectiveness of ECT observed in this study should be interpreted in the context of population-level comparisons rather than severity-specific treatment pathways. Second, the limited availability of relapse and long-term response data for some comparators necessitated making assumptions based on similar treatment classes or extrapolations from the STAR*D study, which may not fully capture real-world treatment dynamics. Third, the model assumed constant treatment effects over time and did not account for potential variations in adherence, discontinuation, or dose adjustments that commonly occur in clinical practice. Moreover, we assumed that within the same Markov health state patients experienced the same health-related quality of life regardless of which treatment led to that state. While this approach is consistent with the way utility values are typically reported and applied in the major depressive disorder literature, it does not capture treatment-specific side effects or treatment burden. In reality, cognitive adverse effects associated with ECT, dissociative symptoms related to esketamine, or the frequent clinic visits required for rTMS may differentially affect patients' quality of life beyond depression symptom status alone. Ignoring such treatment-specific utility decrements may bias QALY estimates in either direction. Future studies incorporating treatment-specific utilities or patient-reported outcome measures would allow more refined comparisons across treatment modalities. Another limitation is that we did not stratify patients based on the degree of prior treatment resistance, which could influence treatment outcomes and potentially overestimate esketamine's effectiveness among more severely resistant cases. Furthermore, our model did not account for sequential treatment scenarios involving switching or progressing through multiple treatment options, particularly the clinical scenario in which esketamine might be used as an effective option after other therapies have failed. Future studies could consider sequential treatment modeling, exploring the potential role of esketamine as a last-resort option following the failure of other more cost-effective therapies, thus better reflecting real-world clinical decision-making. Lastly, IV ketamine was not included as a comparator in this study. Although it is increasingly used in some healthcare settings, it is not routinely incorporated into standardized treatment pathways for TRD within the Hong Kong public healthcare system, and there is limited availability of locally relevant data on its utilization, costs, and real-world effectiveness. While it would be possible to approximate cost inputs using data from other settings, such assumptions would introduce substantial uncertainty and reduce the contextual validity of the model. In addition, its off-label use and heterogeneity in administration protocols present further challenges for robust economic modeling. Given its distinct cost structure—characterized by low drug acquisition costs but relatively

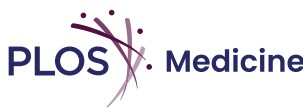

higher administration and service-related costs—its cost-effectiveness is likely to be context-dependent and may vary substantially across healthcare systems. Future research incorporating IV ketamine as a comparator would be valuable in healthcare systems where its use is more established in clinical practice, enabling more context-specific and policy-relevant economic evaluations.

Overall, this study provides a comprehensive real-world–informed economic evaluation of esketamine compared with multiple third-line treatment strategies for TRD. While esketamine was not cost-effective compared with most alternatives under the current WTP threshold, it may offer relative economic advantages over certain neurostimulation treatments such as rTMS and ECT. Importantly, our findings highlight that cost-effectiveness evidence may help inform more context-sensitive treatment sequencing strategies beyond conventional line-of-therapy frameworks. These insights may support reimbursement decision-making, pricing negotiations, and the design of more efficient care delivery models for TRD management.

## Supporting information

**S1 Appendix. Supplemental online content.** Table A. Model input parameters. Table B. Scenario analysis of per patient cost, QALYs and ICER in each comparative arm. Fig A. Schema of retrospective cohort analysis for costs of healthcare resource utilization. Fig B. Deterministic Sensitivity Analysis (DSA) results for ICER of esketamine arm across key model parameters.
(DOCX)

**S2 Appendix. CHEERS 2022 checklist.** Husereau D, Drummond M, Augustovski F, de Bekker-Grob E, Briggs AH, Carswell C, Caulley L, Chaiyakunapruk N, Greenberg D, Loder E, Mauskopf J, Mullins CD, Petrou S, Pwu RF, Staniszewska S; CHEERS 2022 ISPOR Good Research Practices Task Force. Consolidated Health Economic Evaluation Reporting Standards 2022 (CHEERS 2022) Statement: updated Reporting Guidance for Health Economic Evaluations. BMJ. 2022;376:e067975. The checklist is Open Access distributed in accordance with the terms of the Creative Commons Attribution (CC BY 4.0) license, http://creativecommons.org/licenses/by/4.0/.
(DOCX)

## Acknowledgments

The authors would like to thank Zonglin Dai and Deliang Yang for their technical support in model implementation and data processing. We also thank Lisa Lam for her assistance with language editing and proofreading of the manuscript.

## Author contributions

**Conceptualization:** Vivien Kin Yi Chan, Mark Jit, Franco Wing Tak Cheng, Hei Hang Edmund Yiu, Esther Wai Yin Chan, Sandra Sau Man Chan, David Makram Bishai, Dawn Craig.

**Data curation:** Xue Li.

**Formal analysis:** Yifan Li, Vivien Kin Yi Chan.

**Funding acquisition:** Xue Li.

**Methodology:** Yifan Li, Vivien Kin Yi Chan.

**Resources:** Xue Li.

**Software:** Yifan Li.

**Supervision:** Sandra Sau Man Chan, Xue Li.

**Validation:** Vivien Kin Yi Chan.

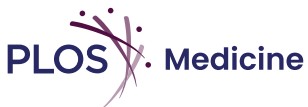

**Writing – original draft:** Yifan Li, Vivien Kin Yi Chan.

**Writing – review & editing:** Mark Jit, Franco Wing Tak Cheng, Hei Hang Edmund Yiu, David Makram Bishai, Dawn Craig, Esther Wai Yin Chan, Sandra Sau Man Chan, Xue Li.

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
