## [Editor Report · Decision Letter 0]

15 Oct 2025

Dear Dr Li,

Thank you for submitting your manuscript entitled "Cost-effectiveness of Esketamine versus Real-world Treatments for Treatment-resistant Depression: A Multi-armed Modelling Study" for consideration by PLOS Medicine.

Your manuscript has now been evaluated by the PLOS Medicine editorial staff, and I am writing to let you know that we would like to send your submission out for external peer review.

For clinical studies, please upload a copy of your trial study protocol as a supporting information file. The study protocol should be the version submitted for approval to the institutional review board or ethics committee, should include any amendments to the study protocol, as well as the date of their approval by the institutional review or ethics committee. Please also detail any deviations from the study protocol in the Methods section of your manuscript. The editors will consider the protocol and study conduct prior to a final decision for external review.

Please re-submit your manuscript within two working days, i.e. by Oct 17 2025 11:59PM.

Kind regards,

Suzanne De Bruijn, PhD

Associate Editor

PLOS Medicine

---

## [Decision Letter · Decision Letter 1]

12 Jan 2026

Dear Dr Li,

Many thanks for submitting your manuscript "Cost-effectiveness of Esketamine versus Real-world Treatments for Treatment-resistant Depression: A Multi-armed Modelling Study" (PMEDICINE-D-25-03549R1) to PLOS Medicine. The paper has been reviewed by subject experts and a statistician; their comments are included below and can also be accessed here: [LINK]

As you will see, your manuscript has been evaluated by three reviewer and an academic editor with relevant expertise. Reviewers stated that this work has potential clinical relevance and is strengthened by the use of real-world comparators. They also asked for further clarifications and raised some concerns about the methodological approach, mainly regarding the comparison of different patient cohorts and the fact that not breaking down costs may limit the generalizability of the findings. You can find a detailed description of the reviews at the end of this letter. Moreover, the Academic Editor noticed that the GitHub repository is not available. Please, ensure that it will be available with your resubmission.

After discussing the paper with the editorial team and an academic editor, I'm pleased to invite you to revise the paper in response to the reviewers' comments. We plan to send the revised paper to some or all of the original reviewers, and we cannot provide any guarantees at this stage regarding publication.

We ask that you submit your revision by Jan 28 2026 11:59PM. However, if this deadline is not feasible, please contact me by email, and we can discuss a suitable alternative.

Don't hesitate to contact me directly with any questions (efourli@plos.org).

Best regards,

Evangelia

Evangelia Fourli,

Associate Editor

PLOS Medicine

efourli@plos.org

Comments from the editor:

Please include Lithium augmentation as a comparator in your analysis.

Comments from the academic editor:

Please ensure that the GitHub repository is available.

Comments from the reviewers:

Reviewer #1: Thank you for the opportunity to review this interesting study of the cost-effectiveness of esketamine for treatment-resistant depression in Hong Kong. The manuscript is particularly innovative for its choice of comparators, which account for widespread use of augmentation therapy beyond oral antidepressants alone. Even with these more costly alternatives in mind, esketamine was found not to be cost-effective at current prices.

Methodologically, this reviewer found the manuscript to be outstanding. The model assumptions are thoughtfully and transparently derived, and eTable1 is exemplary. Analysis of the price reduction required for cost-effectiveness add policy relevance. The use of both deterministic and probabilistic sensitivity analyses further improve the rigor.

In addition to providing the assumed cost per unit of different medications/services consumed (bottom of eTable1), the article could be strengthened by aslo providing the assumptions regarding the number of units consumed at each stage. Please also be sure to provide units for all values. For example, "Treatment Duration" for esketamine is 1 for the induction phase and 5 for the maintenance phase but it is unclear if this is 1 day, 1 week, etc.

Why is the utility weight higher for remission (0.85) than response (0.72)? In the article, "Response" is defined as >= 50% reduction in depression symptom scores, whilst "remission" refers to achieving a score below the response threshold. This suggest higher utility of response. Perhaps this was defined backward?

This reviewer found the schematic in eFigure 1 to be a bit confusing. Do the blue dots signify second-line treatment failure? Why in the first instance is the blue dot labeled as 1st treatment, and the proceeds directly to 3rd treatment? A legend and narrative would be helpful.

Would the authors would be willing to state "in Hong Kong" in the title and abstract, and in the title, instead of "real-world treatments" to use a term such as "standard-of-care" or simply "alternative treatments"?

Please include the CHEERS checklist in the supplementary appendix.

Reviewer #2: PEER REVIEW

Manuscript ID: PMEDICINE-D-25-03549R1

Recommendation: MAJOR REVISION REQUIRED

Summary

This paper addresses an important question—whether esketamine is cost-effective compared to real-world TRD treatments. The attempt to move beyond simple oral antidepressant comparators is valuable, and the use of territory-wide data is a strength. However, fundamental methodological issues undermine confidence in the conclusions. Most critically, treatments are compared using data from different patient populations, key assumptions about treatment effectiveness are unjustified, and the lack of cost disaggregation severely limits generalizability. With substantial revisions, this could make a useful contribution.

MAJOR CONCERNS

1. Treatment comparisons based on studies with different patient populations

The study compares seven different treatments by pulling effectiveness data from different studies and meta-analyses (see eTable 1). The problem is that these studies looked at different patient populations—some more severe, some less treatment-resistant—and used different definitions of what counts as "response" or "remission."

For example, the esketamine trials enrolled patients with a certain level of treatment resistance, the ECT meta-analyses included various severity levels, and the psychotherapy studies may have had different patient characteristics. The authors combine all these effectiveness estimates as if they're directly comparable, but they're not really comparing like with like.

This matters because if esketamine was tested in less severely ill patients than ECT typically treats, or vice versa, then the relative effectiveness estimates could be misleading. When you're making treatment recommendations based on cost-effectiveness, you need confidence that you're making fair comparisons.

Required action: Explain how you accounted for these differences between studies, or acknowledge in the limitations that the comparative effectiveness estimates may be biased because you're combining data from studies with different patient populations. Readers need to know that the head-to-head comparisons aren't based on actual head-to-head trials.

2. Patients with multiple treatment failures assumed to respond like patients with one failure

Lines 197-199 state you assumed third-line, fourth-line and fifth-line treatments have the same effectiveness as STAR*D Level 2, 3, and 4 treatments.

This doesn't make sense clinically. STAR*D participants entered the study after failing ONE antidepressant. Your Level 2 outcomes are from patients with one prior failure. But you're applying those same response rates to patients in their third, fourth, or fifth line of treatment—meaning they've had multiple additional failures.

In clinical practice, we know response rates decline with each failed treatment. A patient who's failed four treatments is fundamentally different from a STAR*D patient who's failed one. Assuming they respond similarly contradicts what we see clinically and likely overestimates how well later-line treatments work.

Required action: Either justify why you think this assumption is valid, or run sensitivity analyses showing what happens if response rates decline with each treatment line (say, 10-20% reduction per line). At minimum, acknowledge this may substantially overestimate effectiveness for all strategies.

3. ECT modeled as if given to average TRD patients, not the severely ill patients who actually receive it

Table 1 shows ECT providing only 0.09 additional QALYs over augmentation therapy over 5 years. This seems too low for the patients who actually get ECT in practice.

In real-world clinical settings, ECT is reserved for the most severe cases—patients with psychotic depression, catatonia, acute suicidality, or who've failed everything else including esketamine. These are not average TRD patients. The model applies average effectiveness from meta-analyses to all TRD patients, but ECT patients are a selected high-severity group where the benefit should be much larger.

Required action: Either model ECT effectiveness for the severity level of patients who actually receive it, or discuss this as a limitation that likely underestimates ECT's value for appropriate candidates.

4. Costs not broken down by type—impossible to apply findings to other countries

Costs are presented as totals without showing how much is drug cost versus labor cost versus equipment. This is a major problem for generalizability because different treatments have completely different cost structures:

Esketamine: mostly drug cost

ECT: mostly labor cost (anesthesiologist, psychiatrist, nurses)

rTMS: mostly labor cost

In Hong Kong with very high labor costs, esketamine looks favorable. But in countries with lower labor costs—most of Asia, Africa, Latin America—ECT and rTMS would likely be much more cost-effective because the labor is cheaper while drug costs stay the same.

Your ECT costs about HKD 9,720 per session (eTable 1). Other published studies have used ECT costs roughly half that. Not coincidentally, those studies found ECT dominant over esketamine, while you find the opposite. The difference is probably driven by labor costs, not by actual clinical differences.

Without breaking down costs, policymakers in other countries can't use your findings. Someone in Vietnam or Kenya can't apply Hong Kong total costs to their setting.

Required action: Show costs broken down into drug costs, labor costs, and equipment costs. Discuss explicitly that these findings apply to high-income settings with high labor costs, and conclusions may be completely different in settings with lower labor costs.

5. I don't understand the Scenario 2 results—can you explain what's happening here?

Looking at eTable 2, when you extend follow-up intervals from every 4 weeks to every 8 weeks (Scenario 2), the incremental QALYs jump from 0.054 to 0.269—that's five times higher.

I don't understand how changing how often you see patients for monitoring would multiply the health benefit by five. Monitoring frequency is an administrative decision, not a treatment efficacy change. Is this a calculation error? Does the longer interval somehow mean treatment continues for longer? Is there some assumption about adherence I'm missing?

This needs a clear explanation because it looks like either something went wrong in the calculations, or the model is behaving in ways that don't make clinical sense.

Required action: Please explain step-by-step why Scenario 2 increases QALYs so dramatically. If it's because treatment duration changes, state that explicitly. If it's an adherence assumption, justify it. If it's a model artifact, acknowledge that.

6. Discussion doesn't address contradictory findings that directly challenge your main conclusions

One of the paper's key findings is that esketamine is more cost-effective than both ECT and rTMS (Table 1, lines 295-297). However, prior published economic evaluations comparing esketamine to ECT have reached the opposite conclusion—finding ECT to be cost-effective or even dominant over esketamine. These studies are in your reference list but the contradiction isn't discussed.

This isn't just minor variation across studies—it's a direct reversal of your main conclusion. When prior research says "ECT wins" and your study says "esketamine wins," readers need to understand why. What's different?

Two factors likely contribute to the opposite findings:

First, cost structure differences: ECT costs vary enormously depending on labor costs (see concern #4). Studies in settings with lower labor costs found ECT favorable, while your Hong Kong setting with very high labor costs finds esketamine favorable.

Second, effectiveness modeling differences: Your model shows ECT producing very minimal QALY gains (see concern #3), while other studies found ECT generated more QALYs than esketamine. If ECT's benefit is underestimated in your model due to not accounting for the high-severity patient population that actually receives it, this would further bias against ECT.

Combined, these two factors—higher modeled costs AND lower modeled effectiveness for ECT—both push toward your "esketamine wins" conclusion. But without discussing this, readers don't know whether your conclusions reflect genuine differences in clinical value or methodological choices in how ECT was modeled.

Required action: Add a paragraph explicitly addressing why your ECT vs. esketamine findings differ from prior studies. Discuss both cost structure differences AND potential differences in how ECT effectiveness was modeled across studies. This is critical because it affects the interpretation of your main finding and its generalizability.

MODERATE CONCERNS

Missing important comparators: Lithium augmentation is a gold-standard treatment that's missing from your analysis. IV ketamine (same mechanism as esketamine, but cheaper) would also be valuable. These omissions limit the claim to evaluate "real-world" options.

Quality of life assumed equal across treatments: Lines 390-391 acknowledge assuming QOL is the same regardless of which treatment patients receive. But ECT has cognitive side effects, esketamine has dissociative effects, and rTMS requires daily visits—these likely affect QOL differently. At minimum, discuss this as a limitation.

SPECIFIC CLARIFICATIONS NEEDED

eTable 1: Many entries say "expert opinion" or "model assumption" without details—please identify the experts or explain the assumptions

eTable 1: "Weighted costs per cycle" calculations aren't shown—please provide the formulas

CONCLUDING COMMENTS

Despite these concerns, the paper has real strengths: moving beyond oral AD monotherapy comparators is important, the multi-scenario approach is thoughtful, and using real-world data enhances relevance.

To be publishable, you need to address: (1) how you handled comparing treatments from different studies, (2) the assumption about treatment line effectiveness, (3) cost disaggregation for generalizability, and (4) discussion of contradictory literature findings.

One broader thought: The field would benefit from more focused comparisons between clinically similar interventions. For example, esketamine versus rTMS—both third-line treatments for similar patients but with opposite cost structures (drug-heavy vs. labor-heavy)—would allow better methodology and more useful findings across different healthcare systems.

Recommendation: Major revision required. The methodological concerns are serious but addressable. With revisions, this could make a useful contribution to the TRD treatment economics literature.

Reviewer #3: This study assesses the cost-effectiveness of intranasal esketamine for treatment-resistant depression (TRD) in Hong Kong using a 5-year Markov cohort model. The authors compare esketamine plus antidepressant therapy with six real-world third-line alternatives spanning pharmacological, psychological, and neurostimulation strategies. A clear strength is the use of realistic comparators rather than antidepressant monotherapy alone, which substantially improves the policy relevance of the analysis. Outcomes are reported in terms of QALYs and healthcare costs, and the authors conclude that esketamine is unlikely to be cost-effective at commonly applied willingness-to-pay thresholds, although its value may improve under alternative scenarios.

The study will be of interest to clinicians, payers, and policymakers making decisions about where esketamine should sit within TRD treatment pathways. The modelling framework and parameterisation are broadly appropriate and consistent with prior economic evaluations in this space, and the results appear internally coherent. That said, several methodological and reporting points need clarification to strengthen confidence in the findings.

Major comments

1) The manuscript states that antidepressant monotherapy was used as a common comparator to support indirect comparisons, and the supplement lists pooled relative effects versus monotherapy. Please clarify the overall evidence synthesis approach used to parameterise relative effectiveness across strategies (i.e. how the pooled effects were used to generate internally consistent inputs across arms), and summarise the key comparability assumptions. A short table listing the main evidence used per comparator (and any key harmonisation steps) would help.

2) The manuscript applies a 2.5% discount rate "based on projected inflation rate", which suggests discounting and price-year adjustment may be conflated. Please clarify how price-year adjustment was handled and how discounting was chosen and applied to both costs and QALYs. For the discount-rate rationale, it would be helpful to cite either a local methods source/HTA precedent for Hong Kong (if available) or, if local guidance is limited, an established international reference case (e.g. NICE uses 3.5% per annum for both costs and health effects). A sensitivity analysis on the discount rate would strengthen the analysis.

3) The scenario analyses focus mainly on esketamine delivery/dosing assumptions. I recommend expanding scenarios to also test key structural/policy assumptions, including:

(i) alternative discounting (e.g. 3.5% per annum for both costs and QALYs as a widely used reference case, such as NICE);

(ii) an explicit price-reduction scenario (e.g. 75%) to directly test the paper's conclusion about the magnitude of reduction needed for cost-effectiveness; and

(iii) longer time horizons (e.g. 10-20 years) to show the impact of truncating the model at 5 years. Even if the 5-year choice is motivated by local approval timelines and limited follow-up, one purpose of modelling is to extrapolate beyond observed data; a longer-horizon scenario would make the implications of this structural choice clearer.

4) Some inputs are informed by expert opinion, but the elicitation process is not described. Please briefly report how these assumptions were derived (e.g. number of experts and how inputs were consolidated) and whether they align with common / good practice. Given the importance of monitoring costs for esketamine, incorporating uncertainty for these parameters in the PSA would strengthen the results.

5) In Figure 3, the PSA scatter for some strategies appears visually bimodal rather than a single compact cloud (notably the upper/green cluster). Please double-check the PSA implementation/sampling logic and add a brief explanation of what drives this shape (e.g. discrete rules/thresholds or other structural features). If unintended, it may indicate an implementation issue.

Minor comments

6) The authors note alignment with CHEERS 2022, but I did not see a completed CHEERS checklist included. Please add a completed CHEERS 2022 checklist as a supplementary file, with page references.

7) The abstract reports a negative ICER for rTMS (e.g. "more cost-effective than rTMS (ICERs: -US$77,400/QALY)"). Negative ICERs are difficult to interpret without stating the quadrant. Here, esketamine appears less costly and more effective than rTMS, so this should be reported as dominance (e.g. "esketamine dominates rTMS"). I suggest replacing the negative ICER wording with dominance language in the abstract and main text.

8) The Discussion states claims like that the results are "highlighting the need for a total price reduction of esketamine of at least 75%" to reach cost-effectiveness, and also notes that decision-makers could consider higher WTP thresholds. I suggest tightening the framing of such claims, so these are presented explicitly as model-based implications under the base-case assumptions (e.g. linearity of drug costs, separation of drug vs monitoring cost components, and the chosen WTP threshold), rather than as general policy conclusions. For example: "Under the base-case assumptions, esketamine would meet a WTP of US$50,000/QALY at approximately a 75% price reduction", and "At higher thresholds, the probability of cost-effectiveness increases, as shown in the CEACs."

---

* Please upload any figures associated with your paper as individual TIF or EPS files with 300dpi resolution at resubmission; please read our figure guidelines for more information on our requirements: http://journals.plos.org/plosmedicine/s/figures. While revising your submission, we strongly recommend that you use PLOS's NAAS tool (https://ngplosjournals.pagemajik.ai/artanalysis) to test your figure files. NAAS can convert your figure files to the TIFF file type and meet basic requirements (such as print size, resolution), or provide you with a report on issues that do not meet our requirements and that NAAS cannot fix.

After uploading your figures to PLOS's NAAS tool - https://ngplosjournals.pagemajik.ai/artanalysis, NAAS will process the files provided and display the results in the "Uploaded Files" section of the page as the processing is complete.

If the uploaded figures meet our requirements (or NAAS is able to fix the files to meet our requirements), the figure will be marked as "fixed" above. If NAAS is unable to fix the files, a red "failed" label will appear above.

When NAAS has confirmed that the figure files meet our requirements, please download the file via the download option, and include these NAAS processed figure files when submitting your revised manuscript.

* Please include the initials of the recipient of the funding (if applicable) and the URL of all funders under the financial disclosure.

FIGURES AND TABLES

SUPPLEMENTARY MATERIAL

REFERENCES

HEALTH ECONOMICS / COST-EFFECTIVENESS STUDIES

* Please ensure that the study is reported according to the CHEERS guideline (available from: https://www.equator-network.org/reporting-guidelines/cheers) and include the completed checklist as Supporting Information. Please add the following statement, or similar, to the Methods: "This study is reported as per the Strengthening the Consolidated Health Economic Evaluation Reporting Standards 2022 (CHEERS 2022) Statement (S1 Checklist)." When completing the checklist, please use section and paragraph numbers, rather than page numbers.

---

## [Decision Letter · Decision Letter 2]

18 Mar 2026

Dear Dr. Li,

Thank you very much for re-submitting your manuscript "Cost-effectiveness of Esketamine versus Alternative Treatment Strategies for Treatment-resistant Depression in Hong Kong: A Multi-armed Modelling Study" (PMEDICINE-D-25-03549R2) for review by PLOS Medicine.

I have discussed the paper with my colleagues and the academic editor and it was also seen again by 3 reviewers. I am pleased to say that provided the remaining editorial and production issues are dealt with we are planning to accept the paper for publication in the journal.

[LINK]

We look forward to receiving the revised manuscript by Mar 24 2026 11:59PM.

Sincerely,

Evangelia Fourli, Ph.D.

Senior Editor

PLOS Medicine

plosmedicine.org

Requests from Editors:

GENERAL EDITORIAL REQUESTS

"* At this stage, we ask that you include a short, non-technical Author Summary of your research to make findings accessible to a wide audience that includes both scientists and non-scientists. The Author Summary should immediately follow the Abstract in your revised manuscript. This text is subject to editorial change and should be distinct from the scientific abstract. Ideally each sub-heading should contain 2-3 single sentence, concise bullet points containing the most salient points from your study. In the final bullet point of ‘What Do These Findings Mean?’ Please include the main limitations of the study in non-technical language.

Please see our author guidelines for more information: https://journals.plos.org/plosmedicine/s/revising-your-manuscript#loc-author-summary."

* Please confirm that your title complies with PLOS Medicine's style. Your title must be nondeclarative and not a question. It should begin with main concept if possible. "Effect of" should be used only if causality can be inferred, i.e., for an RCT. Please place the study design ("A randomized controlled trial," "A retrospective study," "A modelling study," etc.) in the subtitle (ie, after a colon).

* Please confirm that your abstract complies with our requirements, including format (three sections: Background, Methods and Findings, and Conclusions) and providing all the information relevant to this study type https://journals.plos.org/plosmedicine/s/submission-guidelines#loc-abstract

* Please ensure that the Introduction ends with a clear description of the study question or hypothesis.

* Please ensure that all abbreviations are defined at first use throughout the text.

* Please confirm that all numbers presented in the abstract are present and identical to numbers presented in the main manuscript text.

GENERAL

* Please review your text for claims of novelty or primacy (e.g. 'for the first time') and remove this language. In addition, please check that any use of statistical terms (such as trend or significant) are supported by the data, and if not please remove them.

* Please remove the 'conclusions' subheading from the discussion. Please also remove any other subheadings from the discussion.

"* Statistical reporting: Please revise throughout the manuscript, including tables and figures.

- Please report statistical information as follows to improve clarity for the reader ""22% (95% CI [13,28]; p</=)"".

- Please separate upper and lower bounds with commas instead of hyphens as the latter can be confused with reporting of negative values.

- Please repeat statistical definitions (HR, CI etc.) for each set of parentheses."

* In the abstract, please include the important dependent variables that are adjusted for in the analyses.

* In the author summary, please revise formatting and ensure you use bullet points.

* In the author summary, in the final bullet point of 'What Do These Findings Mean?', please include the main limitations of the study in non-technical language.

FUNDING STATEMENT

* The funding statement should include: specific grant numbers, initials of authors who received each award, URLs to sponsors’ websites. Also, please state whether any sponsors or funders (other than the named authors) played any role in study design, data collection and analysis, the decision to publish, or preparation of the manuscript. If they had no role in the research, include this sentence: “The funders had no role in study design, data collection and analysis, decision to publish, or preparation of the manuscript.”

COMPETING INTERESTS STATEMENT

* All authors must declare their relevant competing interests per the PLOS policy, which can be seen here: https://journals.plos.org/plosmedicine/s/competing-interests For authors with ties to industry, please indicate whether any of the interests has a financial stake in the results of the current study.

FIGURES

* Please provide titles and legends for all figures and tables (including those in Supporting Information files). Please define all acronyms used in each figure or table in its corresponding legend.

* Please make sure the formatting of the figures complies with PLOS Medicine guidelines. Specifically:

-For figure files

If you are submitting an Initial or Full Submission and would prefer to embed each figure in the manuscript, do so in read order, immediately following the paragraph where the figure is first mentioned and above the related figure caption.

Upon revision, prepare and submit each figure as an individual file, removing all embedded figures.

-Figure citations

Cite figures in ascending numeric order upon first appearance in the manuscript file.

For detailed instructions, read the guidelines for figures.

-Figure captions

Insert figure captions in the manuscript text, immediately following the paragraph where the figure is first cited (read order). Don’t include captions as part of the figure files themselves or submit them in a separate document.

At a minimum, include the following in your figure captions:

A figure label with Arabic numerals, and “Figure” abbreviated to “Fig” (e.g. Fig 1, Fig 2, Fig 3, etc). Match the label of your figure with the name of the file uploaded at submission (e.g. a figure citation of “Fig 1” must refer to a figure file named “Fig1.tif”).

A concise, descriptive title

The caption may also include a legend as needed.

-These information can be found in the following link: https://journals.plos.org/plosmedicine/s/submission-guidelines#loc-figures-and-tables

*You mentioned that the github link will be available before publication, which is a requirement to publish your work. Can you please include a zenodo link or DOI as well?

*While you indicate that your study uses anonymized data, please include a clear statement regarding informed consent - for example, "the requirement for consent was waived due to the use of fully anonymized data," if applicable, or provide an appropriate alternative statement

Comments from Reviewers:

Reviewer #1: All of Reviewer 1's comments have now been addressed.

Reviewer #2: Thank you for an interesting, clinically useful and very informative article.

I feel that all my comments/questions (and minor objections) have been answered thoroughly, which I am very grateful for.

In the current state I find it of potential use in clinical settings where I have to make treatment choices between a number of options, often with limited evidence to support my decision. Your article is of immediate value to me as a practitioner and will help inform the formation of future local clinical guidelines for treatment resistant depression.

While I actually recommend "accept" in general, I chose option "minor revision" as I would like to make to two further comments:

1) Minor Comment 1: While I understand that the perspective of the article is "currently in Hong Kong", I still would have found the inclusion of IV Ketamine as a treatment option interesting. Perhaps it should be a treatment option in Hong Kong? Or perhaps it should not? Either way, it is extensively used in the US and it might have been possible to model "potential cost" in Hong Kong based on available data.

Suggestion: No change is needed. It is not something that I consider vital to this article. However, if IV ketamine is not added to this article, I would be very interested in reading about the results if you, in future works, added IV Ketamine to your current model to see where a "potential use" of IV Ketamine would fit in regard to cost-effectiveness in Hong Kong.

2) Minor Comment 2: I think that this article highlights an important and often overlooked aspect of treatment options in clinical practice, namely relative cost-effectiveness. If all things are equal, cost effectiveness might inform choice of treatment. In fact, one could argue that third-forth-fifth-line treatment recommendations might be (at least to a degree) guided by cost-effectiveness.

Suggestion: No change is needed. But I think a comment regarding how findings such as the ones in your article may help form more comprehensive guidelines beyond "first-second-third"-line might be warranted. Perhaps a sentence added to conclusions?

Reviewer #3: Thank you for adequately addressing my concerns. I have no further comments.

[LINK]

---

## [Editor Report · Decision Letter 3]

26 Mar 2026

Dear Dr. Li,

Thank you very much for re-submitting your manuscript "Cost-effectiveness of Esketamine versus Alternative Treatment Strategies for Treatment-resistant Depression in Hong Kong: A Multi-armed Modelling Study" (PMEDICINE-D-25-03549R3) for review by PLOS Medicine.

There are a few things that still need to be addressed, which you can find listed at the end of the letter.

We look forward to receiving the revised manuscript by Mar 30 2026 11:59PM.

Sincerely,

Evangelia Fourli, Ph.D.

Senior Editor

PLOS Medicine

plosmedicine.org

Requests from Editors:

- both in the main text and supplementary information, you have references inserted after the punctuation. Can you please revise both files and make sure that references are included before punctuation (including commas, full stops, semi colons etc).

- please, replace arrows with bullet points in the authors summary

- line 10: I suggest removing independently as independent reviews should convey the meaning

- please remove conclusions as an independent text section; it should be included in the discussion. Generally, the discussion should have no subsections

- in the abstract, line 23, you state: " comparing seven third-line strategies". Can I please confirm that this is phrased correctly based on your analysis? In the text it is always mentioned as esketamine vs 6 treatments, so I am wondering if this could be misread. If you believe that this is correct, please disregard this comment.

---

## [Editor Report · Decision Letter 4]

27 Mar 2026

Dear Dr Li,

On behalf of my colleagues and the Academic Editor, Dr Alexander C. Tsai, I am pleased to inform you that we have agreed to publish your manuscript "Cost-effectiveness of Esketamine versus Alternative Treatment Strategies for Treatment-resistant Depression in Hong Kong: A Multi-armed Modelling Study" (PMEDICINE-D-25-03549R4) in PLOS Medicine.

PRESS

Sincerely,

Evangelia Fourli, Ph.D.

Associate Editor

PLOS Medicine